# PgMYB1 Positively Regulates Anthocyanin Accumulation by Activating *PgGSTF6* in Pomegranate

**DOI:** 10.3390/ijms24076366

**Published:** 2023-03-28

**Authors:** Zenghui Wang, Xuemei Yang, Chuanzeng Wang, Lijuan Feng, Yanlei Yin, Jialin Li

**Affiliations:** 1Shandong Institute of Pomology, Tai’an 271000, China; 2School of Biological Science and Technology, University of Jinan, Jinan 250022, China

**Keywords:** anthocyanin, PgMYB1, *PgGSTF6*, pomegranate, transcriptional expression

## Abstract

The peel color of pomegranates is an important exterior quality that determines market value. Anthocyanins are biosynthesized in the cytosol and then transported to the vacuole for storage. However, the molecular mechanism that determines the color variation between red and white pomegranates remains unclear. In this study, we identified an R2R3-MYB protein (PgMYB1) that interacts with the *PgGSTF6* promoter and regulates its transcriptional expression, thus promoting the accumulation of anthocyanins in pomegranate. The expression of *PgMYB1* and *PgGSTF6* was positively correlated with the anthocyanin content in red and white pomegranates. Further investigation showed that the knockdown of *PgMYB1* in red pomegranate ‘Taishanhong’ (TSH), by the virus-induced gene-silencing system, inhibited anthocyanin accumulation. Together, our results indicate that PgMYB1 controls the transport of anthocyanin via *PgGSTF6* and thus promotes anthocyanin accumulation in red pomegranates. Our results have a certain reference value for further clarifying the regulation of anthocyanin synthesis and transport in pomegranate fruits.

## 1. Introduction

Pomegranate (*Punica granatum* L.) originated in central Asian countries [1,2]. In China, many high-genetic-diversity pomegranate varieties have been cultivated for about 2000 years [3]. The exterior fruit color of pomegranates, with red and white variability, reflects differences in anthocyanin concentrations [4]. Anthocyanins play an essential role in seed dispersal, resistance to pathogens, and protection against ultraviolet radiation [5,6,7]. They can also be beneficial for humans in protecting against diseases [8,9,10]. The presence of anthocyanins in pomegranate has important effects on fruit quality; however, the molecular genetics of anthocyanin biosynthesis and transport in pomegranate is still sparse. Many studies have focused on the components and content of pomegranate anthocyanins in fruit peels and juices [11,12].

MYBs are involved in regulating secondary metabolism, organ morphogenesis, and circadian rhythm [13,14,15]. The MYB-bHLH-WD40 (MBW) protein complex regulates anthocyanin biosynthesis [16,17,18]. The transcription levels of anthocyanidin synthase (*ANS*), UDP-glucose:flavonoid-3-O-glucosyltransferase (*UFGT*), and the regulatory gene *PcMYB10* in red pears were found to be significantly higher than those in yellow pears [19]. Glutathione S-transferase (GST) has been shown to play a key role in anthocyanin transport [20,21,22,23,24]. According to sequence relatedness, the GST superfamily can be divided into 14 classes: tau (U), phi (F), lambda (L), theta (T), dehydroascorbate reductase (DHAR), tetrachloro hydroquinone dehalogenase (TCHQD), zeta (Z), glutathionyl hydroquinone reductase (GHR), iota, mPGES-2, Ure2p, elongation factor 1Bγ (EF1Bγ), metaxin, and hemerythrin, of which GSTF is the plant-specific class [25,26,27,28]. In plants, GSTs are also involved in many endogenous biological processes, including flavonoid accumulation and anthocyanin transport [29,30] in various fruit crops, such as in apple, lychee, strawberry, and peach [20,21,31,32,33].

The process of anthocyanin biosynthesis is catalyzed by enzymes in the flavonoid pathway, including chalconesynthase (CHS), chalcone isomerase (CHI), flavanone 3-hydroxylase (F3H), dihydroflavonol 4-reductase (DFR), anthocyanidin synthase (ANS), and UDP-glucose: flavonoid-3-O-glucosyltransferase (UFGT) [34]. MdMYBA binds to the *MdANS* promoter, while MdMYB1 binds to the promoters of *MdDFR*, *MdGSTF6*, and *MdUFGT*. MdMYB10 interacts with MdbHLH3/33 to increase the activity of the *MdDFR* promoter in apple [34,35]. *PgMYB* is important for flavonoid biosynthesis; other transcription factors appear to also be necessary for the regulation of anthocyanin biosynthesis [2]. However, how *PgMYB* regulates *PgGSTF6* expression combined with anthocyanin synthesis and transport in pomegranate is still unknown.

Currently, the mechanisms that determine anthocyanin synthesis and transport in other plants are well known, but little is known specifically regarding pomegranate. In our study, the R2R3-MYB transcription factor PgMYB1 was found to play a key role in anthocyanin accumulation in pomegranate. We hypothesized that PgMYB1 directly activates the expression of *PgGSTF6* and is positively correlated with the expression level of *PgANS*, thus promoting anthocyanin accumulation. We aimed to investigate the key role that PgMYB1 plays in anthocyanin synthesis and transport in pomegranate.

## 2. Results

### 2.1. Phylogenetic Analysis and Subcellular Localization of PgMYB1

To identify the putative *MYB1* gene in the pomegranate genome, BlastP was used to search the pomegranate genome database based on apple MYB1 (NP_001288045.1) protein. To explore the homology of the *PgMYB1* gene among different plant species, a phylogenetic tree was constructed. We found that PgMYB1 was clustered within the R2R3-MYB clade, which includes MdMYB1 (apple), CmMYB6 (chrysanthemum), and other proteins that have been identified as being involved in anthocyanin synthesis (Figure 1a). The comparative analysis of the amino acid sequence of PgMYB1 and its close homologs showed that the PgMYB1 protein is an R2R3-MYB transcription factor (Appendix A). We constructed a translational fusion protein, *35S:PgMYB1*-GFP, to determine the subcellular localization of PgMYB1. The *35S:PgMYB1*-GFP fusion protein showed a fluorescent signal in the nucleus (Figure 1b,c), suggesting that PgMYB1 was translated in the nucleus. 

### 2.2. Identification and Analysis of GSTF Genes in Pomegranate

To identify the putative GSTF family genes in the pomegranate genome, BlastP was used to search against the pomegranate genome database based on 15 apple GSTF proteins. A total of 18 *GSTF* genes were identified in the pomegranate genome. The comparative analysis of the amino acid sequences of 18 GSTF proteins and their close homologs showed that the pomegranate GSTF proteins are highly conserved (Appendix A). To explore the homology of *GSTF* genes among different plant species, a phylogenetic analysis was conducted to compare GSTF proteins from pomegranate, *Arabidopsis*, apple, dragon’s blood tree, and maize. PgGSTF6 was clustered within the GSTF clade, which includes MdGSTF6 and other proteins identified as being involved in anthocyanin transport (Figure 2), suggesting that PgGSTF6 may be involved in pomegranate anthocyanin transport.

### 2.3. Expression Patterns of PgMYB1, PgANS, and PgGSTFs in Different Developmental Stages of TSH and SBT

Previous research has shown that *MdMYB1* plays a key role in anthocyanin transport via *MdGSTF6* [34]. To further confirm that *PgMYB1* is involved in anthocyanin accumulation via *PgGSTFs* in pomegranate, samples of two pomegranate cultivars, TSH, with red fruit peels, and ‘Sanbaitian’ (SBT), with white fruit peels at maturity, were screened as experimental materials (Figure 3a). We observed higher expression levels of *PgMYB1* (GenBank accession number: OWM81764.1) in S1–S4 of TSH than those in SBT (Figure 3b). The expression level trends of *PgGSTF9* (GenBank accession number: NC_045127.1) and other *PgGSTF* genes were not obvious in different developmental stages of TSH and SBT (Figure 3b and Appendix A). We also observed lower expression levels of *PgANS* (GenBank accession number: AHZ97874.1) and *PgGSTF6* (GenBank accession number: NC_045131.1) in S1–S6 of SBT than in TSH, indicating that *PgMYB1*, *PgANS*, and *PgGSTF6* play a key role in anthocyanin synthesis and transport in pomegranate (Figure 3b). This finding prompted us to explore the mechanism leading to high anthocyanin levels in red pomegranate.

### 2.4. PgMYB1 Positively Regulates Anthocyanin Accumulation in Pomegranate

Previous studies have clarified that MdMYB1 positively regulates anthocyanin accumulation in apple [34]. To demonstrate whether PgMYB1 is essential for pomegranate fruit coloration, we used a virus-induced gene-silencing system to conduct a transient expression assay. The injection sites on the pomegranate peel turned lighter red after transformation with pTRV2-*PgMYB1* and pTRV1. However, there was no obvious phenotypic change observed after the transformation with the pTRV2 vector (Figure 4a). Additionally, the anthocyanin content in pTRV2-*PgMYB1* pomegranate peels was lower than that in pTRV2 pomegranate peels (Figure 4b). 

Knockdown of *PgMYB1* strongly downregulated the expression of *PgMYB1*. Furthermore, we performed qRT-PCR to further study the expression patterns of other genes in the anthocyanin biosynthesis pathway in pTRV2-*PgMYB1* pomegranate peels. We observed that *PgGSTF6* and *PgANS* were less expressed in pTRV2-*PgMYB1* pomegranate peels (Figure 4c). Knockdown of *PgMYB1* affected the expression level of *PgGSTF6* and *PgANS.* Together, our results show that the lower expression of *PgMYB1* may affect anthocyanin accumulation via downregulated expression of *PgGSTF6* and *PgANS*, indicating that *PgMYB1* plays a key role in anthocyanin accumulation in pomegranate.

### 2.5. PgMYB1 Directly Regulates PgGSTF6 Transcription

To further understand the mechanism by which *PgMYB1* regulates anthocyanin accumulation in pomegranate, yeast one-hybrid (Y1H) assays were performed. These assays showed that PgMYB1 binds to the *PgGSTF6* promoter and activates its expression (Figure 5a). To further investigate whether PgMYB1 activates the transcriptional expression of *PgGSTF6*, transient expression assays were conducted. As expected, the *PgMYB1* plus *proPgGSTF6:GUS* treatments showed significantly higher GUS activity (Figure 5b,c). Furthermore, we performed qRT-PCR to further study the expression level of *GUS* in tobacco leaves. We found higher *GUS* expression levels in *PgMYB1* plus *proPgGSTF6:GUS* tobacco leaves than in the *proPgGSTF6:GUS* leaves (Figure 5d). These results show that PgMYB1 could bind to the *PgGSTF6* promoter and activate its expression in pomegranate. 

To further reveal how the transcriptional regulation mechanism of PgMYB1 regulates the *PgGSTF6* gene expression, the promoter sequence of *PgGSTF6* was analyzed. We observed that there was a potential MBS binding site (CTGTTG) in the *PgGSTF6* gene promoter region (−638–633) (Figure 5e and Appendix A). To further determine if PgMYB1 could directly bind to the binding site, yeast one-hybrid (Y1H) assays were performed. As expected, PgMYB1 could interact with MBS (CTGTTG), while PgMYB1 could not bind when the MBS (CTGTTG) site was mutated to mMBS (CGGTGG) (Figure 5f). Together, these data indicate that PgMYB1 directly binds to the CTGTTG motif of the *PgGSTF6* promoter region and activates its expression.

## 3. Discussion

Fruit color is one of the most important agronomic traits of fruit quality and is an important attribute in terms of consumer preference. Anthocyanins in fruit not only confer different bright colors but are also beneficial for human health. Therefore, it is of great significance to study the regulation of anthocyanin biosynthetic pathways. Glutathione S-transferase (GST) has been shown to play a key role in anthocyanin transport [20,21,22,23,24].

Previous studies have shown that the abundance of GSTs in several horticultural crops, including peach, lychee, grape, and strawberry, is related to the pigmentation of the fruits [20,21,33,36]. *MdGSTF6* is highly expressed in red apple fruits, and its expression abundance is positively correlated with anthocyanin content [34]. To date, the transcriptional regulation mechanism of GSTs in pomegranate has not been well investigated. In this study, we identified that *PgGSTF6*, a *GST* gene, was clustered within the GSTF clade, including MdGSTF6, which is positively correlated with anthocyanin content and is highly expressed in red apple fruits (Figure 2). We found a lower expression level of *PgGSTF6* in S1–S6 of SBT than that in TSH, indicating that *PgGSTF6* plays a key role in anthocyanin transport in pomegranate (Figure 3b). This finding provides new insight into anthocyanin transport and prompted us to explore the mechanism leading to high anthocyanin levels in red pomegranates.

*MdMYB1* is an anthocyanin accumulation gene that responds to light signals in apple [37]. We found that PgMYB1 was clustered within the R2R3-MYB clade, including MdMYB1 and CmMYB6 (Figure 1a), which are involved in anthocyanin accumulation. Subcellular localization analysis of PgMYB1 showed that the *35S:PgMYB1*-GFP fluoresced in the nucleus (Figure 1b,c), suggesting that PgMYB1 may be involved in pomegranate anthocyanin accumulation as a transcription factor. However, how PgMYB1 regulates the process of anthocyanin synthesis and transport in pomegranate needs further exploration.

In apple, MdMYBA and MdMYB1 regulate anthocyanin biosynthesis in fruit peel. Additionally, MdMYBA binds to the *MdANS* promoter, while MdMYB1 activates the transcription of *MdDFR* and *MdUFGT* [34,35]. We found that the anthocyanin content in pTRV2-*PgMYB1* pomegranate peels was lower than that in pTRV2 pomegranate peels. Previous studies have shown that the *GSTs* play key roles in anthocyanin accumulation in *Arabidopsis*, petunia, and maize [38,39,40]. In our study, the lower expression levels of *PgANS* in peels of white fruits suggest that the lack of *PgANS* expression may be the main factor responsible for the absence of anthocyanins in white pomegranate. Furthermore, knockdown of *PgMYB1* strongly downregulated the expression of *PgGSTF6* and *PgANS* but did not affect the transcription of the other genes in the anthocyanin biosynthesis pathway (Figure 4c). We found that PgMYB1 could activate the expression of *PgGSTF6* in pomegranate (Figure 5). Together, these results show that knockdown of *PgMYB1* affects anthocyanin accumulation via downregulated expression of *PgGSTF6* and *PgANS.*

Anthocyanin accumulation in fruit is a key quality trait in pomegranate. However, there have been few studies on anthocyanin accumulation in pomegranate. Therefore, how anthocyanin accumulation is regulated is important for clarifying the mechanism underlying pomegranate fruit coloration. In this study, our results reveal that the lower expression of *PgMYB1* may affect anthocyanin accumulation via downregulated expression of *PgGSTF6* and *PgANS* in pomegranate. This is because PgMYB1 directly activates the expression of the glutathione S-transferase (*GST*) gene *PgGSTF6* and affects the expression level of the encoding anthocyanidin synthase (*ANS*) gene, *PgANS*, thus promoting anthocyanin accumulation (Figure 6). Our results reveal the mechanism leading to anthocyanidin accumulation in red peel pomegranate and provide new insights that are useful in the cultivation of red peel pomegranate with high contents of anthocyanins, which are beneficial to human health.

## 4. Materials and Methods

### 4.1. Plant Materials and Growth Conditions

The two pomegranate cultivars, TSH, with red fruit peels, and SBT, with white fruit peels at maturity, were grown in Tai’an, Shandong Province, China. Fruit peels were collected on the following six sampling dates, 10 July, 25 July, 9 August, 24 August, 8 September, and 23 September, and were designated as S1, S2, S3, S4, S5, and S6, respectively. We collected nine fruits from each cultivar at each stage. Peels from these fruits were pared with a knife and then immediately frozen in liquid nitrogen and stored at −80 °C until use. 

### 4.2. RNA Extraction and qRT-PCR Analysis 

Total RNA was extracted from ‘TSH’ and ‘SBT’ pomegranate peels using a Total RNA Isolation System (Tiangen, Beijing, China). The RT-PCR reactions were performed using qPCR SuperMix (TransGen Biotech, Beijing, China) with an Applied Biosystems 7500 real-time PCR system (Applied Biosystems, New York, NY, USA). Expression data were normalized to those of the pomegranate *actin* gene. Transcription levels were calculated using the cycle threshold (Ct) 2^−ΔΔCt^ method, according to Livak et al. (2001) [41]. All primers used in this study are listed in Appendix A.

### 4.3. Sequence Alignments and Phylogenetic Analysis

The sequence alignment was performed as described by Wang et al. (2019) [13]. Two phylogenetic trees were constructed using the full-length amino acid sequences of 18 pomegranate, 15 apple, 14 *Arabidopsis*, 5 dragon’s blood tree, and 7 maize GSTF protein sequences and 8 R2R3-MYB transcription factor protein sequences using MEGA 7.0 and the neighbor-joining (NJ) method as described by Li et al. (2022) [15].

### 4.4. Yeast One-Hybrid Assays

According to the manufacturer’s instructions, Y187 (Clontech, Palo Alto, CA, USA) was used for Y1H assays. The full-length *PgMYB1* CDS was cloned into the pGADT7 vector to generate the construct AD-PgMYB1. The 2000 bp promoter sequences of PgGSTF6 were inserted into the pHIS2 vector (Clontech, Palo Alto, CA, USA) to generate the construct pHIS2–proPgGSTF6. AD-PgMYB1 and pHIS2–proPgGSTF6 were co-transformed into yeast Y187. The negative control was the pHIS2 empty vector. All primers used in this study are listed in Appendix A.

### 4.5. Transient Expression Assays

Transient expression assays were conducted using tobacco leaves. The 2000 bp promoter sequences of *PgGSTF6* were cloned into pCAMBIA1300 (Takara, Dalian, China) to generate the construct *proPgGSTF6:GUS*. The CDS of *PgMYB1* was inserted into pCAMBIA1300 to generate the construct *35S:PgMYB1*. The different combinations were co-transformed into tobacco leaves via an *agrobacterium*-mediated method. The injected tobacco grew for 2–3 days under normal conditions. The *GUS* activity was detected via GUS staining and *GUS* expression quantity. All primers used in this study are listed in Appendix A.

### 4.6. Virus-Induced Gene Silencing of PgMYB1 in Pomegranate

A 459 bp fragment of PgMYB1 was amplified and cloned into the pTRV2 vector to generate the construct pTRV2-PgMYB1, and then the pTRV2, pTRV2-PgMYB1, and pTRV1 were each transformed into the GV3101 *A. tumefaciens* strain for the next VIGS experiments. The pTRV2 empty vector was used as the control. The pTRV2-PgMYB1 and pTRV2 vectors were injected into pomegranate peel along with pTRV1. The fruits of the ‘TSH’ cultivar were selected to inject the VIGS constructs. A total of 80 fruits were selected and divided into two groups, with one to three injections for each single fruit. The expression analysis was conducted on the infected area. The VIGS analysis was performed as described by Jiang et al. (2019) [34]. All primers used for VIGS experiments are listed in Appendix A.

### 4.7. Measurement of Total Anthocyanins

The total anthocyanins of the fruit peels of TSH fruits after *PgMYB1* silencing and pTRV2 empty vector were extracted using the HCl–methanol method. Then, 0.5 g of each sample was fully ground in liquid nitrogen and incubated with 5 mL cold 1% (v/v) methanol–hydrochloric acid in the dark for 24 h. The absorbances of samples at 650, 620, and 530 nm were measured using a UV-1600 spectrophotometer (Shimadzu, Kyoto, Japan). We used the formula OD=(A530–A620)–0.1(OD650–OD620) to calculate the anthocyanin contents of fruit peels of TSH fruits after *PgMYB1* silencing and pTRV2 empty vector control treatment, according to Li et al. (2020) [42].

### 4.8. Subcellular Localization Analyses

PgMYB1 was cloned into the pRI101 vector (Takara, Dalian, China) to construct the 35S:PgMYB1-GFP recombinant vector. The vector was then transformed into *Agrobacterium tumefaciens* LBA4404 competent cells. Transgenic onion epidermal cells of 35S:PgMYB1-GFP were obtained as described by Wang et al. (2017) [43], while 35S:GFP was used as a negative control. All primers used in this study are listed in Appendix A.

## Figures and Tables

**Figure 1 ijms-24-06366-f001:**
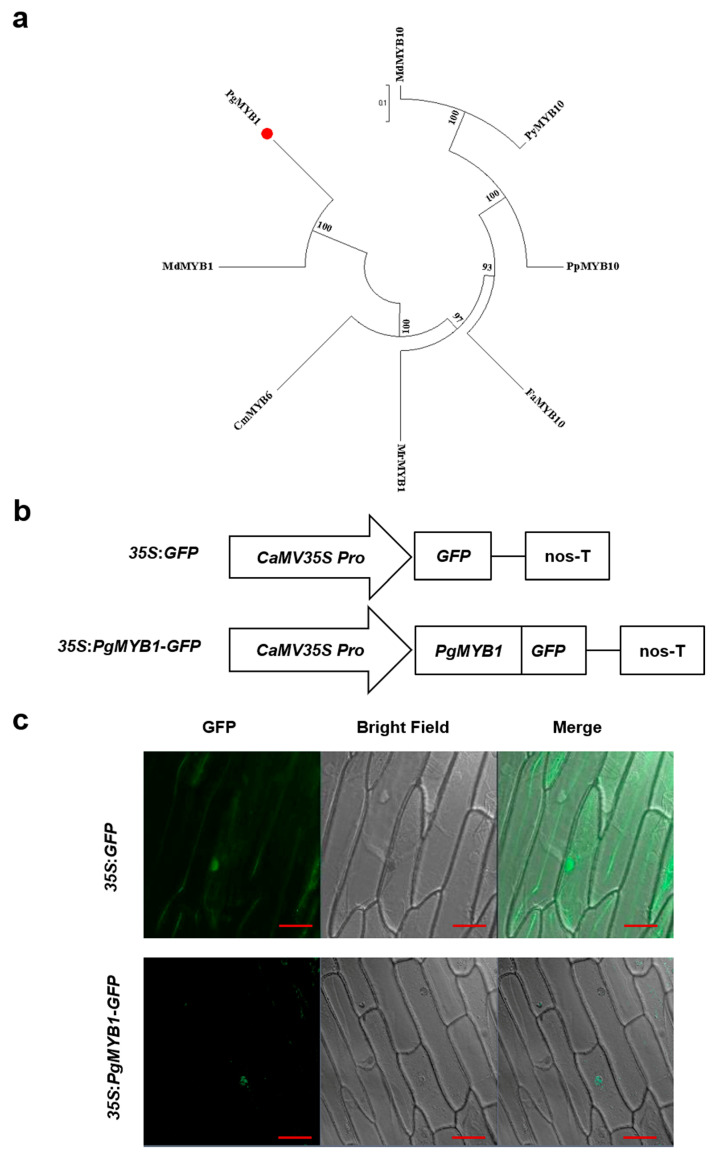
Phylogenetic analysis and subcellular localization of PgMYB1 protein in onion epidermal cells. (**a**) Phylogenetic tree of *PgMYB1* and its close homologs in plant species. The red circle represents PgMYB1. (**b**) Schematic illustration of control construct (*35S:GFP*) and fusion vector (*35S:PgMYB1-GFP*). (**c**) PgMYB1 is localized in the nucleus of onion epidermal cells. GFP, green fluorescent protein. Bars, 50 μm.

**Figure 2 ijms-24-06366-f002:**
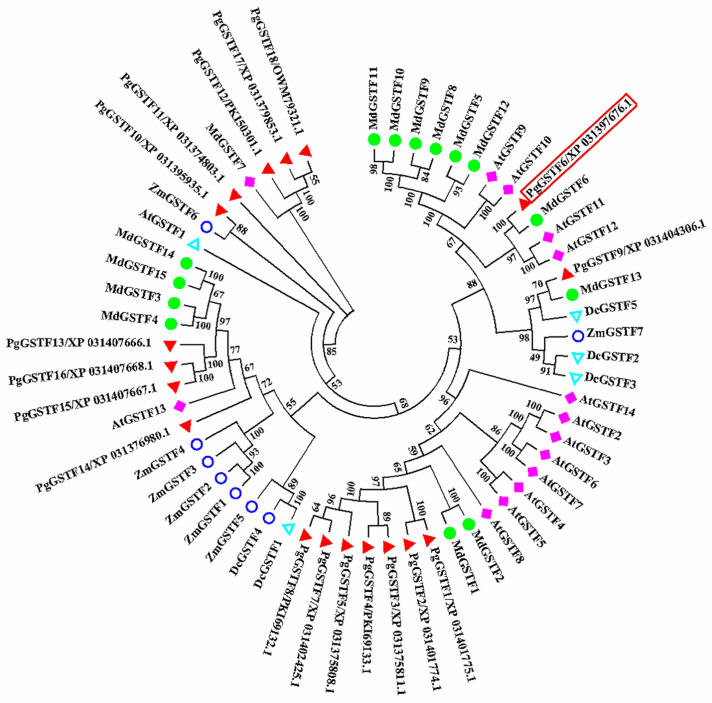
Phylogenetic tree of GSTF proteins in pomegranate, *Arabidopsis*, apple, dragon’s blood tree, and maize. The 18 pomegranate, 15 apple, 14 *Arabidopsis*, 5 dragon’s blood tree, and 7 maize GSTF protein sequences were aligned using ClustalW. The long red rectangular box represents PgGSTF6. The phylogenetic tree was constructed based on the neighbor-joining algorithm.

**Figure 3 ijms-24-06366-f003:**
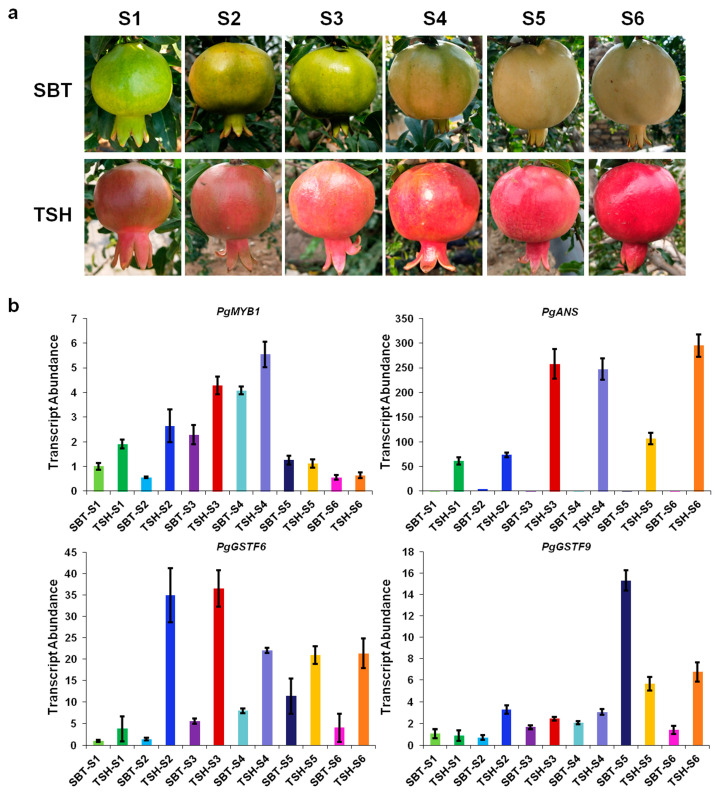
Expression patterns of *PgMYB1*, *PgANS*, and *PgGSTFs* in different developmental stages of TSH and SBT. (**a**) Stages of development in TSH and SBT pomegranates. The stages are designated as S1, S2, S3, S4, S5, and S6. (**b**) The expression levels of *PgMYB1*, *PgANS*, *PgGSTF6*, and *PgGSTF9* in the different stages of development in TSH and SBT fruits. qRT-PCR was conducted in three biological replicates and three technical replicates, and one representative biological repeat is shown.

**Figure 4 ijms-24-06366-f004:**
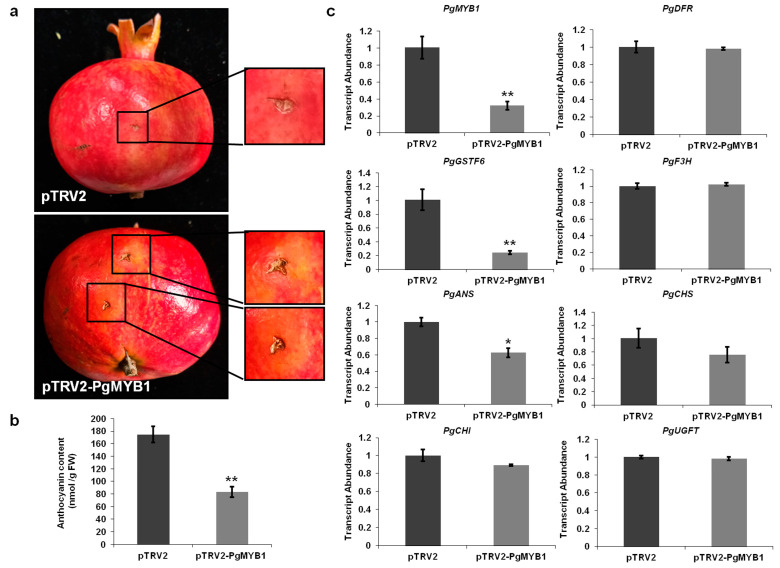
PgMYB1 positively regulates anthocyanin accumulation in pomegranate. (**a**) The phenotypes of TSH fruits after *PgMYB1* silencing. The pTRV2 empty vector was used as the control. (**b**) The anthocyanin contents in fruit peels of TSH fruits after *PgMYB1* silencing. ** *p*-values < 0.01 were considered statistically significant. (**c**) Transcription levels of *PgMYB1*, *PgGSTF6*, and the genes in the anthocyanin pathway in TSH fruits after *PgMYB1* silencing and the pTRV2 empty vector control. Chalconesynthase (CHS accession number: AHZ97870.1), chalcone isomerase (CHI accession number: AHZ97871.1), flavanone 3-hydroxylase (F3H accession number: AHZ97872.1), dihydroflavonol 4-reductase (DFR accession number: AHZ97873.1), anthocyanidin synthase (ANS accession number: AHZ97874.1), and UDP-glucose: flavonoid-3-O-glucosyltransferase (UFGT accession number: AHZ97875.1). qRT-PCR was conducted in three biological replicates and three technical replicates, and one representative biological repeat is shown. * *p*-values < 0.05, ** *p*-values < 0.01 were considered statistically significant.

**Figure 5 ijms-24-06366-f005:**
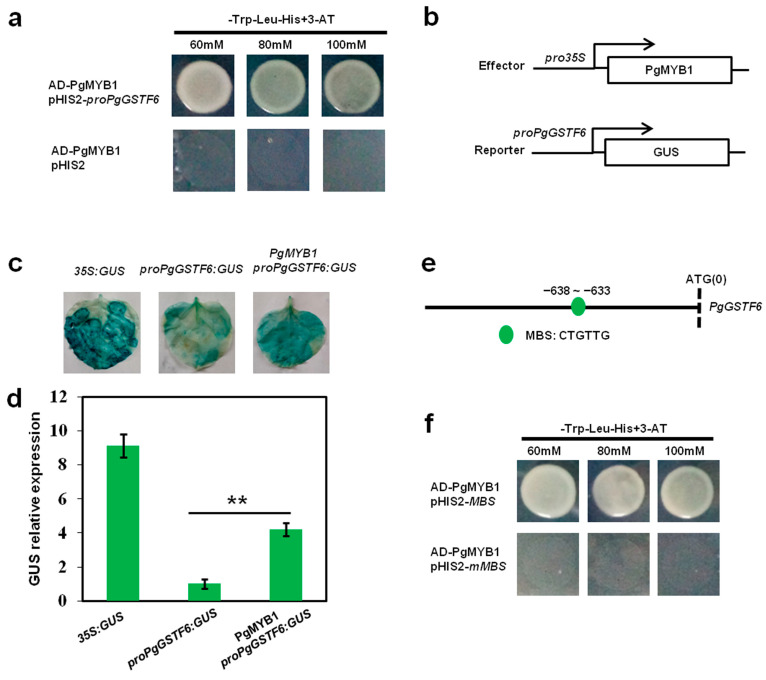
PgMYB1 directly regulates *PgGSTF6* transcription. (**a**) Yeast-one-hybrid assays showed that PgMYB1 binds to the *PgGSTF6* gene promoter. Positive colonies were screened on SD/-Trp-Leu-His medium containing optimal concentrations of 60 mM, 80 mM, and 100 mM 3-AT. (**b**) The *35S:PgMYB1* effector construct and the *proPgGSTF6:GUS* reporter construct were used for activity assays. (**c**) The *PgMYB1* plus *proPgGSTF6:GUS* leaves showed significantly higher *GUS* activity. (**d**) qRT-PCR showed a higher *GUS* expression level in *PgMYB1* plus *proPgGSTF6:GUS* tobacco leaves than in *proPgGSTF6:GUS* leaves. qRT-PCR was conducted in three biological replicates and three technical replicates, and one representative biological repeat is shown. ** *p*-values < 0.01 were considered statistically significant. (**e**) The location of the MBS (CTGTTG) motif on the *PgGSTF6* gene promoter. (**f**) Yeast-one-hybrid assays showed that PgMYB1 binds to the MBS (CTGTTG) motif of the *PgGSTF6* gene promoter, but not to the mMBS (CGGTGG). Positive colonies were screened on SD/-Trp-Leu-His medium containing optimal concentrations of 60 mM, 80 mM, and 100 mM 3-AT.

**Figure 6 ijms-24-06366-f006:**
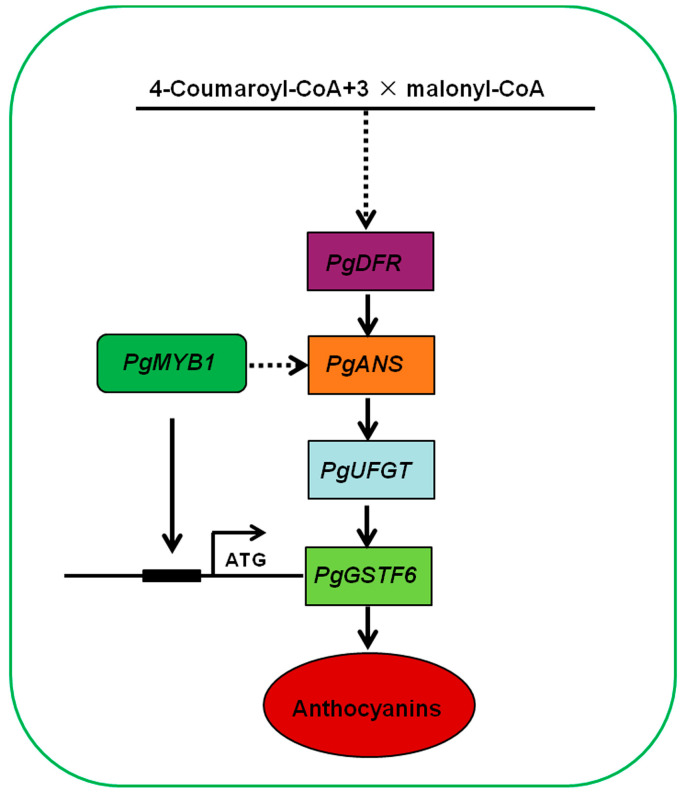
A working model of PgMYB1 function in anthocyanin accumulation. PgMYB1 directly binds to the promoter of *PgGSTF6*, thereby activating its transcriptional expression. Additionally, knockdown of *PgMYB1* downregulated the expression of *PgANS*, indicating that *PgMYB1* may activate transcriptional expression of *PgANS* and thus affect anthocyanin accumulation in pomegranate.

## Data Availability

Data are contained within the article or Appendix A.

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
