# Peer review of "PgMYB1 Positively Regulates Anthocyanin Accumulation by Activating PgGSTF6 in Pomegranate"

_ijms, 2023, doi:10.3390/ijms24076366_

Round 1

Reviewer 1 Report (Previous Reviewer 2)

I have not additional requests

Author Response

Comments and Suggestions for Authors:

I have not additional requests.

Response: Thank you very much for your warmly work.

Reviewer 2 Report (Previous Reviewer 1)

The article 'PgMYB1 positively regulates anthocyanin accumulation by activating PgGSTF6 in pomegranate' by Zenghui Wang et al. has undergone a significant revision following the previous suggestions, and I believe it is looking better now. Have no further comments.

Author Response

Comments and Suggestions for Authors

The article 'PgMYB1 positively regulates anthocyanin accumulation by activating PgGSTF6 in pomegranate' by Zenghui Wang et al. has undergone a significant revision following the previous suggestions, and I believe it is looking better now. Have no further comments.

Response: Thank you very much for your warmly work.

This manuscript is a resubmission of an earlier submission. The following is a list of the peer review reports and author responses from that submission.

Round 1

Reviewer 1 Report

Dear Authors

The manuscript entitled "PgMYB1 positively regulates anthocyanin accumulation by activating PgGSTF6 in pomegranate" is exciting and may aid in improving pomegranate consumer preference and value. However, I have some suggestions for authors.

The writing needs to be improved to make it more beneficial to the scientific community. I have found many issues with the English usage and description. The paper as well ought to be reported using the past tense. I have given a few below in the attached file. I implore the authors not to limit themselves to only outline issues but to check the entire manuscript.

Thank you.

Reviewer 2 Report

This study explores the role of PgMYB1 and PgGSTF6 in anthocyanin accumulation in pomegranate. Given the form of the manuscript, it is difficult to provide a correct evaluation of the scientific merit of the work. The manuscript needs a major revision of the language and style, a critical review by a scientific editor would help immensely.

Some comments

1)     “2.1. Phylogenetic analysis and subcellular localization of PgMYB1”

It is unclear how the PgMYB1 sequence was selected for this study: previous studies or a blast search with a MYB1 sequence of other plant species?

The subcellular localization of the 35S:PgMYB1-GFP fusion protein does not imply any functional correlation with anthocyanin biosynthesis. I believe that many transcriptional factors are expressed in the nucleus. Possible assumptions can only be made within the discussion, considering the data as a whole.

2)    “2.3. Identification and analysis of GSTF Genes in pomegranate”

I think that the rationale should be: I hypothesize that PgGSTFs are involved in anthocyanin biosynthesis, I perform a phylogenetic analysis of PgGSTFs, and I choose, for subsequent analysis, those that cluster with proteins already characterized and involved in anthocyanin biosynthesis. If I’m correct, the real time PCR analysis (paragraph 2.2) should follow this part. However, without a clear explanation of this part its meaning is vague.

3)    2.4. PgMYB1 positively regulates anthocyanin accumulation in pomegranate

In this part several genes were tested by real time PCR (Fig 4c). The authors conclude that “PgMYB1 is positively correlated with the expression level of PgGSTF6 and PgANS” (135-6). What about the other genes that are negatively correlated?

This should be discussed somewhere.

Was the expression analysis conducted on the whole fruit peel or only on the infected area? I can see an effect of the PgMYB1 silencing (fig. 4a) but this seems to be localized. In addition, the control fruit also shows some parts that are less pigmented.

It is unclear the number of samples used for this analysis. It is also unclear the number of infections for each single fruit. This experiment needs a good description.  

Anthocyanin content (Fig. 4b) is expressed as nmol/gFW. Is it correct?  Jiang paper (ref.34) used the same formula for anthocyanin quantification and the content is expressed as mg/gFW. Please specify.  

Moreover, in the materials and methods is reported that: “The total anthocyanins of the different developmental stages peels of TSH and SBT were extracted …..” (275-276). I do not see these data in the manuscript

4)    In general materials and methods should be better explained

5)    Figure 6. Description of the model should be improved. Why is a cloud marked by light?

6)    Discussion should be improved.

Author Response

Reviewer #2:

Comment 1: “2.1. Phylogenetic analysis and subcellular localization of PgMYB1”

It is unclear how the PgMYB1 sequence was selected for this study: previous studies or a blast search with a MYB1 sequence of other plant species?

The subcellular localization of the 35S:PgMYB1-GFP fusion protein does not imply any functional correlation with anthocyanin biosynthesis. I believe that many transcriptional factors are expressed in the nucleus. Possible assumptions can only be made within the discussion, considering the data as a whole.

Response: Thank you very much for your suggestion. We added the method of selecting PgMYB1 sequence in the revised manuscript (lines 75-76).

In our study, subcellular localization analysis showed that PgMYB1 was a transcription factor. PgMYB1 was clustered within the R2R3-MYB clade including MdMYB1 and CmMYB6 (Figure 1a), which are involved in anthocyanin accumulation, indicating that PgMYB1 may be involved in pomegranate anthocyanin accumulation. Furthermore, we added the discussion of subcellular localization of the 35S:PgMYB1-GFP fusion protein in the section of “3. Discussion” (lines 266-267).

Comment 2: “2.3. Identification and analysis of GSTF Genes in pomegranate”

I think that the rationale should be: I hypothesize that PgGSTFs are involved in anthocyanin biosynthesis, I perform a phylogenetic analysis of PgGSTFs, and I choose, for subsequent analysis, those that cluster with proteins already characterized and involved in anthocyanin biosynthesis. If I’m correct, the real time PCR analysis (paragraph 2.2) should follow this part. However, without a clear explanation of this part its meaning is vague.

Response: According to your suggestion, we adjusted the logical structure of this part in our revised manuscript.

Comment 3: 2.4. PgMYB1 positively regulates anthocyanin accumulation in pomegranate

In this part several genes were tested by real time PCR (Fig 4c). The authors conclude that “PgMYB1 is positively correlated with the expression level of PgGSTF6 and PgANS” (135-6). What about the other genes that are negatively correlated?

This should be discussed somewhere.

Was the expression analysis conducted on the whole fruit peel or only on the infected area? I can see an effect of the PgMYB1 silencing (fig. 4a) but this seems to be localized. In addition, the control fruit also shows some parts that are less pigmented.

It is unclear the number of samples used for this analysis. It is also unclear the number of infections for each single fruit. This experiment needs a good description. 

Anthocyanin content (Fig. 4b) is expressed as nmol/gFW. Is it correct?  Jiang paper (ref.34) used the same formula for anthocyanin quantification and the content is expressed as mg/gFW. Please specify. 

Moreover, in the materials and methods is reported that: “The total anthocyanins of the different developmental stages peels of TSH and SBT were extracted …..” (275-276). I do not see these data in the manuscript

Response: In our study, knockdown of PgMYB1 strongly downregulated the expression of PgGSTF6 and PgANS but did not affect the transcription of the other genes in the anthocyanin biosynthesis pathway. We discussed this in the section of “3. Discussion” (lines 276-278).

Thank you very much for your suggestion. The expression analysis was conducted on the infected area.

A total of 80 fruits were selected and divided into two groups and 1-3 of infections for each single fruit. In the section of “4.6. Virus-induced gene silencing of PgMYB1 in pomegranate”, a comprehensive and detailed description of this experiment was added (Lines 340-341).

We confirmed that anthocyanin content (Fig. 4b) is expressed as nmol/gFW was accurated, according to Li et al. (2020). Li, J.; Luan, Q.; Han, J.; Zhang, C.; Liu, M.; Ren, Z. CsMYB60 directly and indirectly activates structural genes to promote the biosynthesis of flavonols and proanthocyanidins in cucumber. Hortic Res.2020, 7, 103.

Thank you very much for your suggestion. We corrected this part in our revised manuscript (lines 345-346).

Comment 4: In general materials and methods should be better explained

Response: Considering your suggestion, we added a detailed description about some parts of materials and methods in our revised manuscript (Lines 300-302; 337-341; 345-346; 352-353).

Comment 5: Figure 6. Description of the model should be improved. Why is a cloud marked by light?

Response: Considering your suggestion, we improved the model and deleted a cloud marked by light in Figure 6.

Comment 6: Discussion should be improved.

Response: According to your suggestion, we adjusted and improved part of the discussion in our revised manuscript.

Round 2

Reviewer 2 Report

This new version of the manuscript is indeed improved but some work is still needed. The are many issues that need to be fixed.  Please, pay attention to the repetition of concepts.

Some comments

Lines 12-13 “In this study, we identified an R2R3-MYB protein (PgMYB1), caused it to interact with PgGSTF6, and regulated its transcriptional expression in pomegranate.”

Consider changing it in:

In this study, we identified an R2R3-MYB protein (PgMYB1) that interacts with the PgGSTF6 promoter and regulates its transcriptional expression in pomegranate.

Line 15, “TSH” The acronyms should be specified the first time they are used not later in the text.

Lines 17-19 These sentences are repetitive. Moreover, a similar concept is already expressed in lines 12-13.

Lines 25-26  “In  China, many high-genetic-diversity pomegranate plants have been cultivated about 2000 years [3].”

Consider changing it in:

In China, many high-genetic-diversity pomegranate varieties have been cultivated for about 2000 years [3].”

Lines 29-31 repeat similar concepts  

Line 37 The acronyms ANS, UFGT must be specified

Lines 44-48 “In plants, GSTs are also involved in many endogenous biological processes, including flavonoid accumulation and anthocyanin transport [29,30]. Genes from this class are involved in the anthocyanin transport process in various fruit crops, such as MdGST in apple, LcGST4 in lychee, RAP in strawberry, and Riant in peach [20,21,31-33].

Consider changing it in:

In plants, GSTs are also involved in many endogenous biological processes, including flavonoid accumulation and anthocyanin transport [29,30] in various fruit crops, such as in apple, lychee, strawberry, and peach [20,21,31-33].

Lines 52-53 “At the transcriptional level, the MYB-bHLH-WD40 protein complex regulates anthocyanin biosynthesis in plants.” and lines 36-37   “The MYB-bHLH-WD40 (MBW) protein complex regulates anthocyanin biosynthesis [16-18]. “ express the same concept.

 Line 70, please specify the id of the apple MYB1 protein used for BLASTP analysis

 Lines 70-71, “To further identify how PgMYB1 is involved in anthocyanin accumulation in pomegranate, a phylogenetic tree was constructed.”

This sentence is incorrect. A phylogenetic analysis cannot explain how PgMYB1 is involved in anthocyanin accumulation.

This phylogenetic analysis only show that PgMYB1 clusters with some proteins that are involved in anthocyanin accumulation. In my opinion this figure does not provide any special information as only a few number of sequences have been used. The phylogenetic analysis of figure 2 makes more sense.

Line 79, “suggesting that PgMYB1 may be involved in pomegranate anthocyanin accumulation.”

The analysis only shows that PgMYB1 is translated in the nucleus. This hypothesis can only be done in the discussion when all the data produced are considered.

I do not see the description of this experiment in the materials and methods

 Figure 1C The quality of figure must be improved.

 Figure 6, the model is unclear. The model suggest that PgMYB1 activates both the red and white pigmentations ??????

 qRT-PCR analysis,  I do see the list of the primers used for the analysis.

 Discussion must be improved.

Author Response

Reviewer #2:

Comment 1: Lines 12-13 “In this study, we identified an R2R3-MYB protein (PgMYB1), caused it to interact with PgGSTF6, and regulated its transcriptional expression in pomegranate.”

Consider changing it in:

In this study, we identified an R2R3-MYB protein (PgMYB1) that interacts with the PgGSTF6 promoter and regulates its transcriptional expression in pomegranate.“2.1. Phylogenetic analysis and subcellular localization of PgMYB1”

Response: Thank you very much for your suggestion. We changed it in our revised manuscript.

Comment 2: Line 15, “TSH” The acronyms should be specified the first time they are used not later in the text.

Response: According to your suggestion, a detailed description of ‘TSH’ was added in our revised manuscript (line 18).

Comment 3: Lines 17-19 These sentences are repetitive. Moreover, a similar concept is already expressed in lines 12-13.

Response: According to your suggestion, we adjusted the logical structure of this part in our revised manuscript (lines 15-16; 19-20).

Comment 4: Lines 25-26  “In  China, many high-genetic-diversity pomegranate plants have been cultivated about 2000 years [3].”

Consider changing it in:

In China, many high-genetic-diversity pomegranate varieties have been cultivated for about 2000 years [3].”

Response: Thank you very much for your suggestion. We changed it in our revised manuscript.

Comment 5: Lines 29-31 repeat similar concepts 

Response: Considering your suggestion, we deleted the repeat similar concepts in our revised manuscript (line 35).

Comment 6: Line 37 The acronyms ANS, UFGT must be specified

Response: According to your suggestion, a detailed description of ANS, UFGT was added in our revised manuscript (lines 43-44).

Comment 7: Lines 44-48 “In plants, GSTs are also involved in many endogenous biological processes, including flavonoid accumulation and anthocyanin transport [29,30]. Genes from this class are involved in the anthocyanin transport process in various fruit crops, such as MdGST in apple, LcGST4 in lychee, RAP in strawberry, and Riant in peach [20,21,31-33].

Consider changing it in:

In plants, GSTs are also involved in many endogenous biological processes, including flavonoid accumulation and anthocyanin transport [29,30] in various fruit crops, such as in apple, lychee, strawberry, and peach [20,21,31-33].

Response: Thank you very much for your suggestion. We changed it in our revised manuscript.

Comment 8: Lines 52-53 “At the transcriptional level, the MYB-bHLH-WD40 protein complex regulates anthocyanin biosynthesis in plants.” and lines 36-37   “The MYB-bHLH-WD40 (MBW) protein complex regulates anthocyanin biosynthesis [16-18]. “ express the same concept.

Response: Considering your suggestion, we deleted the repeat similar concepts in our revised manuscript (lines 62-63).

Comment 9: Line 70, please specify the id of the apple MYB1 protein used for BLASTP analysis

Response: According to your suggestion, the id of the apple MYB1 protein was added in our revised manuscript (line 82).

Comment 10: Lines 70-71, “To further identify how PgMYB1 is involved in anthocyanin accumulation in pomegranate, a phylogenetic tree was constructed.”

This sentence is incorrect. A phylogenetic analysis cannot explain how PgMYB1 is involved in anthocyanin accumulation.

This phylogenetic analysis only show that PgMYB1 clusters with some proteins that are involved in anthocyanin accumulation. In my opinion this figure does not provide any special information as only a few number of sequences have been used. The phylogenetic analysis of figure 2 makes more sense.

Response: According to your suggestion, we modified this sentence in our revised manuscript (lines 83-84).

Thank you very much for your suggestion. The phylogenetic analysis showed that PgMYB1 was clustered within the R2R3-MYB clade including MdMYB1 and CmMYB6, which are involved in anthocyanin accumulation, indicating that PgMYB1 may be involved in pomegranate anthocyanin accumulation. In addition, the main focus of Figure 1a was to preliminarily verify whether PgMYB1 was highly similar to MYB protein sequences of other species involved in anthocyanin synthesis.

Comment 11: Line 79, “suggesting that PgMYB1 may be involved in pomegranate anthocyanin accumulation.”

The analysis only shows that PgMYB1 is translated in the nucleus. This hypothesis can only be done in the discussion when all the data produced are considered.

I do not see the description of this experiment in the materials and methods

Response: Thank you very much for your suggestion. We modified this sentence in our revised manuscript (lines 94-95).

Considering your suggestion, we added a detailed description of subcellular localization analyses in the part of materials and methods in our revised manuscript (lines 374-379).

Comment 12: Figure 1C The quality of figure must be improved.

Response: According to your suggestion, we improved the quality of Figure 1C in our revised manuscript.

Comment 13: Figure 6, the model is unclear. The model suggest that PgMYB1 activates both the red and white pigmentations ??????

Response: The model suggested that PgMYB1 activated the red pomegranate pigmentations and we modified the model in our revised manuscript.

Comment 14: qRT-PCR analysis,  I do see the list of the primers used for the analysis.

Response: All primers used of qRT-PCR analysis are listed in Table S1.

Comment 15: Discussion must be improved.

Response: According to your suggestion, we improved part of the discussion in our revised manuscript (lines286-289; 291-292; 295-298).

Many thanks to you for your good comments.

Round 3

Reviewer 2 Report

Additional comments

Line 13  that interacts with the PgGSTF6 promoter and regulated its transcriptional expression, thus pro..

I would change in:  that interacts with the PgGSTF6 promoter and regulates its transcriptional expression, thus pro...

Fig.1c  35S:PgMYB1-GFP, the GFP signal in this figure is hard to see in my pdf version

Line 94 analysis was conducted to compare GSTF proteins from pomegranate, A. thaliana, apple, dragon’s blood tree, and maize,

I would change A. thaliana in Arabidopsis to be coherent

Line 96  includes MdGSTF6 and others which have been identified as being involved in antho…

I would change it in: MdGSTF6 and other proteins identified as being involved in antho…

Figure 2. be coherent with arabidopsis nomenclature

Lines 212-218 MdMYB1 is an anthocyanin accumulation gene that responses to light signals in  apple [37]. We found that PgMYB1 was clustered within the R2R3-MYB clade, including MdMYB1 and CmMYB6 (Figure 1a), which are involved in anthocyanin accumulation, indicating that PgMYB1 may regulate anthocyanin accumulation in pomegranate. Subcellular localization analysis of PgMYB1 showed that the 35S:PgMYB1-GFP fluoresced in 216 the nucleus (Figure 1b,c), suggesting that PgMYB1 may be involved in pomegranate anthocyanin accumulation as a transcription factor.

To avoid repetitions, I would change it in:

MdMYB1 is an anthocyanin accumulation gene that responses to light signals in  apple [37]. We found that PgMYB1 was clustered within the R2R3-MYB clade, including MdMYB1 and CmMYB6 (Figure 1a), which are involved in anthocyanin accumulation. Subcellular localization analysis of PgMYB1 showed that the 35S:PgMYB1-GFP fluoresced in the nucleus (Figure 1b,c), suggesting that PgMYB1 may be involved in pomegranate anthocyanin accumulation as a transcription factor.

In general, acronyms need not be specified multiple times, Figure 3 legend is an example.

Figure 6 legend. “PgMYB1 directly activates the expression of PgGSTF6 and is positively correlated with the expression level of PgANS” . The figure suggests that PgMYB1 directly activates both PgGSTF6 and PgANS.

Author Response

Reviewer #2:

Comment 1: Line 13  that interacts with the PgGSTF6 promoter and regulated its transcriptional expression, thus pro..

I would change in:  that interacts with the PgGSTF6 promoter and regulates its transcriptional expression, thus pro...

Response: Thank you very much for your suggestion. We changed it in our revised manuscript.

Comment 2: Fig.1c  35S:PgMYB1-GFP, the GFP signal in this figure is hard to see in my pdf version

Response: According to your suggestion, We changed it in a new figure (35S:PgMYB1-GFP) in our revised manuscript.

Comment 3: Line 94 analysis was conducted to compare GSTF proteins from pomegranate, A. thaliana, apple, dragon’s blood tree, and maize,

I would change A. thaliana in Arabidopsis to be coherent

Response: We changed it in our revised manuscript.

Comment 4: Line 96  includes MdGSTF6 and others which have been identified as being involved in antho…

I would change it in: MdGSTF6 and other proteins identified as being involved in antho…

Response: We changed it in our revised manuscript.

Comment 5: Figure 2. be coherent with arabidopsis nomenclature

Response: Corrected as suggested.

Comment 6: Lines 212-218 MdMYB1 is an anthocyanin accumulation gene that responses to light signals in  apple [37]. We found that PgMYB1 was clustered within the R2R3-MYB clade, including MdMYB1 and CmMYB6 (Figure 1a), which are involved in anthocyanin accumulation, indicating that PgMYB1 may regulate anthocyanin accumulation in pomegranate. Subcellular localization analysis of PgMYB1 showed that the 35S:PgMYB1-GFP fluoresced in 216 the nucleus (Figure 1b,c), suggesting that PgMYB1 may be involved in pomegranate anthocyanin accumulation as a transcription factor.

To avoid repetitions, I would change it in:

MdMYB1 is an anthocyanin accumulation gene that responses to light signals in  apple [37]. We found that PgMYB1 was clustered within the R2R3-MYB clade, including MdMYB1 and CmMYB6 (Figure 1a), which are involved in anthocyanin accumulation. Subcellular localization analysis of PgMYB1 showed that the 35S:PgMYB1-GFP fluoresced in the nucleus (Figure 1b,c), suggesting that PgMYB1 may be involved in pomegranate anthocyanin accumulation as a transcription factor.

Response: We changed it in our revised manuscript.

Comment 7: In general, acronyms need not be specified multiple times, Figure 3 legend is an example.

Response: Corrected as suggested.

Comment 8: Figure 6 legend. “PgMYB1 directly activates the expression of PgGSTF6 and is positively correlated with the expression level of PgANS” . The figure suggests that PgMYB1 directly activates both PgGSTF6 and PgANS.

Response: Considering your suggestion, we modified the model in our revised manuscript.

Many thanks to you for your good comments.
